# A Lightweight and High-Precision Passion Fruit YOLO Detection Model for Deployment in Embedded Devices

**DOI:** 10.3390/s24154942

**Published:** 2024-07-30

**Authors:** Qiyan Sun, Pengbo Li, Chentao He, Qiming Song, Jierui Chen, Xiangzeng Kong, Zhicong Luo

**Affiliations:** 1College of Computer and Information Sciences, Fujian Agriculture and Forestry University, Fuzhou 350100, China; sunqiyan99168@163.com; 2College of Mechanical and Electrical Engineering, Fujian Agriculture and Forestry University, Fuzhou 350100, China; lipengbo2526@163.com (P.L.); xzkong@fafu.edu.cn (X.K.); 3College of Jinshan, Fujian Agriculture and Forestry University, Fuzhou 350100, China; c200110228875117@163.com

**Keywords:** passion fruit detection, lightweight, deep learning, knowledge distillation, embedded devices

## Abstract

In order to shorten detection times and improve average precision in embedded devices, a lightweight and high-accuracy model is proposed to detect passion fruit in complex environments (e.g., with backlighting, occlusion, overlap, sun, cloud, or rain). First, replacing the backbone network of YOLOv5 with a lightweight GhostNet model reduces the number of parameters and computational complexity while improving the detection speed. Second, a new feature branch is added to the backbone network and the feature fusion layer in the neck network is reconstructed to effectively combine the lower- and higher-level features, which improves the accuracy of the model while maintaining its lightweight nature. Finally, a knowledge distillation method is used to transfer knowledge from the more capable teacher model to the less capable student model, significantly improving the detection accuracy. The improved model is denoted as G-YOLO-NK. The average accuracy of the G-YOLO-NK network is 96.00%, which is 1.00% higher than that of the original YOLOv5s model. Furthermore, the model size is 7.14 MB, half that of the original model, and its real-time detection frame rate is 11.25 FPS when implemented on the Jetson Nano. The proposed model is found to outperform state-of-the-art models in terms of average precision and detection performance. The present work provides an effective model for real-time detection of passion fruit in complex orchard scenes, offering valuable technical support for the development of orchard picking robots and greatly improving the intelligence level of orchards.

## 1. Introduction

Passion fruit and its byproducts are highly nutritious and have significant commercial value that can be exploited [1]. Passion fruit cultivation is mainly distributed in regions such as Guangdong, Yunnan, and Fujian in China, among others. The planting area is expanding, and the number of varieties is increasing. At present, passion fruit picking is still mainly carried out by hand, which undoubtedly requires a great deal of labor. The development of agricultural robotic picking is of great significance in terms of liberating labor and leading the fruit industry towards a precision model [2]. In recent years, the use of image technology to detect fruits has garnered research interest and emerged as a prominent topic determining the accuracy and integrity of agricultural robotic picking efforts.

Traditional machine learning approaches are based primarily on manually designed combinations of features and classifiers [3]. For example, the basic texture, color, and shape features of fruits have been studied. Tu et al. [4] established an RGB color space model to detect the maturity of passion fruit. Li et al. [5] used a region classifier to classify ripe and unripe tomatoes and used the Hough transform circle detection method to achieve detection of unripe tomatoes; however, this takes a long time and does not have high detection accuracy. Yang et al. [6] attempted to use various machine learning methods to classify apricots based on their shape features. The above image recognition methods suffer from poor robustness and difficulty in handling large volumes of data. Object detection technology mainly involves identifying and classifying the positions to be detected in images or videos. There are several algorithms for target detection, which can be generally classified into those based on two-stage detection, such as Faster R-CNN [7], and those based on one-stage detection, such as SSD [8] and YOLO [9]. Notably, one-stage detection algorithms have higher detection speed, which is beneficial for mobile deployment.

The YOLO algorithm, with its simple structure and short inference time, is one of the best choices for detection models [10]. Lawal et al. [11] combined DenseNet with the YOLOv3 network and used the Mish activation function to detect tomatoes. Roy et al. [12] added DenseNet and SPP blocks, and improved PANet to YOLOv4 to enhance the network’s detection capability. Lin et al. [13] improved the YOLOv4 network by incorporating attention mechanisms, with the goal of eliminating noise and enhancing the feature extraction of small targets. In addition, the reliability and accuracy of the model were enhanced by using the point–line distance loss function [14] and optimizing the upsampling algorithm [15] to improve the YOLOv5 model. Although the above study improved the detection accuracy of the model, it increased the number of parameters and detection time. Researchers have conducted various studies seeking to compress and accelerate models in terms of various aspects [16], including lightweight network designs, pruning, and knowledge distillation. Lightweight network design methods are used to design small models and quickly recognize networks by adjusting their internal structure, with examples including MobileNet [17], GhostNet [18], ShuffleNet [19], and more. The purpose of pruning is the same as lightweight network design, involving the removal of redundant parameters from the network through techniques such as channel pruning [20], kernel pruning [21,22], and weight pruning [23]. The knowledge distillation method proposes to transfer information from one model to another, which efficiently extracts features and can substantially improve detection accuracy [24,25].

Deploying deep learning models for agricultural detection on mobile devices is more meaningful for practical applications [26]. Researchers have effectively improved different models to improve detection accuracy and recognition speed. Xu et al. [27] introduced GhostNet to replace the YOLOv4 backbone network and an effective channel attention mechanism in the neck to detect fruit. The size of the improved model was 43.5 MB, and the detection time for a single image was 48.2 ms. Jiang et al. [28] proposed Generalized-FPN (GFPN), with a cross-scale connection structure integrating the features of the previous and current layers. Subsequently, Xu et al. [29] improved GFPN and applied it to the YOLO network, increasing its accuracy by 1.4%. Guo et al. [30] introduced a knowledge distillation strategy using the YOLOv5s model and achieved an accuracy of 94.67% on a self-constructed dataset, which was 4.83% higher than the original model. Yang et al. [31] constructed a lightweight model based on the backbone replacement, sparse training, and knowledge distillation techniques, which reduced the number of parameters and model size; however, the AP also decreased by 2.7%. Although the above methods have made progress in reducing model weight and enhancing accuracy compared to the original models, the balance between accuracy and weight has not yet been fully realized. Therefore, it is important to study detection algorithms with high generalization ability for use in embedded devices.

For this study, we constructed a passion fruit dataset taken in a complex environment and aimed to address the issues of parameter redundancy and poor real-time model performance on embedded devices. This study presents the G-YOLO-NK model, which is a lightweight and high-precision model based on an improved YOLOv5. The first contribution of this study is that we used a lightweight GhostNet to replace the YOLOv5s backbone in order reduce the number of parameters and computational complexity of the network compared to other methods that use a lightweight network as the backbone network. Second, we reconstructed the neck of the network by combining the new branches of the feature extraction layer with the feature fusion layer. Finally, we used the knowledge distillation method to enable student models to learn useful knowledge from teacher models, verifying the effectiveness of using the distillation method with a one-stage detector. The experimental results show that the improved algorithm reduces the number of model parameters while improving the detection speed, and has better real-time detection performance in complex environments on embedded devices.

The remainder of this article is organized as follows. In Section 2, the materials and methods related to preprocessing image datasets and detection algorithms are presented; Section 3 describes the training methods and evaluation metrics; Section 4 provides the results of the study and comparison experiments; and Section 5 presents the conclusions and outlook for future research.

## 2. Materials and Methods

### 2.1. Image Acquisition and Preprocessing

In order to enhance the single passion fruit dataset, data collection was conducted at the Junzhiyu Passion Fruit Base in Minhou County, Fuzhou City, Fujian Province. The image acquisition device was a Nikon digital camera, the distance of the camera from the passion fruit was 80–100 cm during acquisition, and the image size was 1920 × 1080 pixels, with the images saved in JPEG format. The weather at the time of data collection included sunny, rain, and cloudy conditions. Images of passion fruit were captured under different lighting conditions and compositions in order to enhance the diversity of the dataset. This included toplight, backlight, leaf shading, and fruit overlap scenarios.

There were 3269 images in total, including 837 unshaded fruits, 1124 shaded by leaves, and 1308 overlapping fruit, of which 1732 were normally toplight and 1537 were backlight. Examples of the collected images are shown in Figure 1.

Enhancing images can emphasize the overall or local features of passion fruit images, enhance the differences between different object features, and suppress the extraction of irrelevant features by deep learning networks [32]. Expanding the image training set is advantageous in improving the learning capacity of deep neural networks and reducing overfitting caused by insufficient sample diversity [33]. As such, this approach has the potential to greatly enhance the robustness and generalization capabilities of the trained model. Therefore, we expanded the dataset through the use of image enhancement techniques such as rotating the original data, adding Gaussian noise, and contrast adjustment. Examples of the enhanced images are shown in Figure 2. Rotating the original image by 90 degrees, adding Gaussian noise, and adjusting the contrast were carried out to increase the recognition ability of the model, as shown in Figure 2b–d. After the aforementioned offline data augmentation process, 5140 images of passion fruit were finally obtained. The above images were manually marked and bounding boxes were drawn using LabelImg software, version 3.4.1, which eventually generated xml format files. The completed dataset was randomly divided into training and test sets [34] using an 8:2 ratio, resulting in 4112 images allocated for training and 1028 images for testing.

### 2.2. YOLOv5 Algorithm

The YOLOv5 target detection algorithm was released by Ultralytics in 2020. It has high accuracy and fast inference ability, making it one of the best-performing target detection models available at present [35]. The YOLOv5 model can be separated into four parts: Input, Backbone, Neck, and Detector. The input uses Mosaic data enhancement to randomly scale, cut, and stitch the passion fruit images into the network, which not only enriches the dataset but also enhances the robustness of the network model. The backbone network adds Focus, C3, and SPP structures to the YOLOv3 network. The main role of the backbone network is to extract the features of the image and enhance the learning ability of the convolutional neural network. The path aggregation network (PANet) structure is applied in the neck network, which effectively extracts comprehensive location information from top to bottom while simultaneously capturing semantic features from bottom to top. This integration enhances the localization of targets by leveraging both spatial and semantic information. The detection network produces the final output by combining the probability class of the target, the confidence score, and the location information of the target box. The structure of the YOLOv5 algorithm is shown in Figure 3.

Researchers have developed four different YOLOv5 models based on varying the depth and width of the network, demonstrating the exceptional flexibility of this algorithm. This indicates that the algorithm is highly adaptable and can be customized to suit different requirements. In this study, the detection performance of the four models was tested on the constructed passion fruit dataset. Table 1 shows the test results. In order to save memory on embedded devices, the YOLOv5s model was chosen as the baseline in this study. The overall loss of YOLOv5s encompasses the classification, localization, and confidence losses. The cross-entropy loss function was employed for the classification and confidence losses, simplifying the computational complexity, while the CIoU Loss was used for localization loss, which helps to ensure that the model can accurately locate the target.

### 2.3. Improvement of the YOLOv5s Model

With the aim of reducing the model’s size and computational complexity while improving its detection accuracy, we propose various improvements to the YOLOv5s model. The structure of the improved YOLOv5s algorithm is shown in Figure 4. As evident from the figure, substituting the backbone network of YOLOv5 with a streamlined GhostNet model along with the introduction of a fresh feature branch within the backbone and the subsequent reconfiguration of the feature fusion layer in the neck network yielded remarkable results. These enhancements not only elevate the model’s accuracy but also amplify its detection speed, presenting a superior balance between performance and efficiency.

#### 2.3.1. Lightweight Improvements

Due to the limited storage space and computing resources of embedded devices, deploying deep learning models can be quite challenging [36], requiring further model compression [37]. GhostNet has been shown to outperform MobileNet and ShuffleNet in terms of providing computational performance with a compact network design [38]. A model with outstanding performance has sufficient complexity in the feature layer for understanding the input information, which is an important factor in the success of a model [39]. In lightweight network design, it is not feasible to simply remove useful redundant features; therefore, GhostNet was specifically designed to enable fast inference on mobile devices while maintaining important features. The Ghost module in GhostNet is the key structure for generating feature layers, which facilitates the extraction of effective feature layers. The Ghost module, shown in Figure 5, uses a series of inexpensive linear operations to generate new feature layers, which may be 1 × 1 or 3 × 3 convolutions.

Suppose that the input channel is denoted by *c*, the height and width of the feature map by *h* and *w*, respectively, the height and width of the output feature map by *h*′ and *w*′, respectively, the number of convolution kernels by *n*, the size of the convolution kernel by *k*, the size of the linear transform convolution kernel by *d*, and the number of transforms by *s*; then, the parameter compression using Ghost convolution instead of conventional convolution is shown in Equation (Equation 1). The acceleration ratio is derived as shown in Equation (Equation 2).
(1)$$r_{c}=\frac{n\cdot c\cdot k\cdot k}{\frac{n}{s}\cdot c\cdot k+\left(s-1\right)\cdot \frac{n}{s}\cdot d\cdot d}\approx \frac{s\cdot c}{s+c-1}\approx s$$
(2)$$r_{c}=\frac{n\times h^{\prime }\times w^{\prime }\times c\times k\times k}{\frac{n}{s}\times h^{\prime }\times w^{\prime }\times c\times k\times k+\left(s-1\right)\times h^{\prime }\times w^{\prime }\times d\times d}\approx \frac{s\times c}{s+c-1}\approx s$$

It can be observed from the equations that the benefits of computational acceleration and parameter compression are influenced by the number of transformations. In the Ghost module, the total number of parameters and the computational complexity are reduced compared to a normal convolutional neural network without changing the size of the output feature layer. The overall structure of GhostNet as the backbone network is detailed in Table 2.

#### 2.3.2. Reconstructing the Neck Network

In deep learning networks, while the robustness and generalization ability of an improved model depends on modifying the backbone network, modifying the neck network can have a similar effect. In order to identify objects at different scales, Adelson et al. [40] first proposed using the image pyramid to build a feature pyramid, which has since been applied to image analysis, data compression, and image processing. However, this approach calculates features at each image scale slowly and inaccurately. To address this problem, the top-down connected Feature Pyramid Network (FPN) [41] and Path Aggregation Network (PANet) [42] have been proposed to boost the information flow. Jiang et al. [28] proposed Generalized-FPN for efficient object detection, which improves FPN with a novel queen-fusion approach. In order to achieve the goal of multiscale information exchange, in this study we propose an adaptive feature pyramid network (AFPN) based on the Generalized-FPN idea in order to effectively detect passion fruit targets.

After replacing the YOLOv5 backbone with GhostNet, the information in the feature extraction layer is reduced, resulting in a loss of information in the neck feature fusion layer and a consequent decrease in the detection performance of the model. At the same time, it can be seen from the images in the dataset that most passion fruits are medium and large targets; therefore, we first connect the 80 × 80 × 64, 40 × 40 × 80, and 40 × 40 × 112 feature layers of the backbone network to the fusion layer of the neck, enhancing the feature extraction ability for medium targets and compensating for the lost information in the lightweight network. Second, we add a branch in the 40 × 40 × 256 feature layer of the neck and convert it to the scale of 20 × 20 × 128 through convolutional operations to connect with the detection layer, enhancing the network’s ability to detect large targets. Finally, we used an image for detection to verify the detection capability of the improved network and the effectiveness of the improved method. Figure 6 shows feature images selected from the 27th layer of the G-YOLOv5 and G-YOLOv5-N models, from which it can be seen that G-YOLOv5 lacks semantic and positional information, while the improved G-YOLOv5-N compensates for this drawback. With the above modification scheme, the neck network possesses strong semantic features at a high level and localization features at a low level. The structure of the adaptive feature pyramid network (AFPN) is shown in Figure 7.

#### 2.3.3. Knowledge Distillation Enhancement

Knowledge distillation (KD) is an effective method for further improving a model’s detection accuracy [43]. Distillation is not yet widely used in the YOLO series of network improvements, especially for small single-target models. We conducted a special study for G-YOLOv5-N and used the distillation technique to achieve a final further improvement to the detection effect of the G-YOLOv5-N model. First, the teacher network model was chosen rationally. We chose the YOLOv5 series of models in order to ensure that the student and teacher models use the same scale in the output layer. Next, the YOLOv5x model was selected as the teacher model based on its high accuracy during training on the passion fruit dataset.

In general, the implementation of distillation involves parameter initialization, training a teacher network, then using the rich knowledge learned by the teacher network to train a student network. A flowchart detailing the use of the knowledge distillation algorithm is shown in Figure 8.

Teacher networks can be used for predictive learning in student networks. To enhance the information exchange between them, the predictions of the teacher network are used as soft labels. The teacher network then trains the student network using these soft labels, allowing the student network to learn from the teacher’s knowledge. Additionally, the student network helps to prevent the teacher network from making mistakes through learning from hard labels. By incorporating soft labels, the student network can acquire more nuanced and hidden knowledge. This hidden knowledge is usually expressed as a categorical output $y_{i}$′, while $y_{i}$ represents the input knowledge or information, as shown in Equation (Equation 3): (3)$${y_{i}}^{\prime }=\frac{\exp y_{i}}{\sum \exp y_{i}}.$$
From Equation (Equation 3), it can be determined that the model does not facilitate learning of the dark knowledge in the passion fruit images. Thus, a warming process is needed, as detailed in Equation (Equation 4), where *T* represents the temperature coefficient: (4)$${y_{i}}^{\prime }=\frac{\exp(y_{i}/T)}{\sum \exp(y_{i}/T)}.$$

The cumulative loss function utilized in the knowledge distillation algorithm comprises the original network model’s loss and the distillation loss. The distillation loss is composed of the classification, bounding box, and localization losses. To promote the model’s learning of passion fruit targets, the background region is weakened by introducing a weighting factor *K*. The distillation loss is detailed in Equation (Equation 5), while the total loss equation is shown in Equation (Equation 6).
(5)$$L_{dloss}=K\left(L_{obj}+L_{cl}+L_{bb}\right)$$
(6)$$L_{loss}=\alpha L_{dloss}+\left(1-\alpha \right)L_{yolo}$$

## 3. Model Training and Evaluation

### 3.1. Experimental Environment

In order to comprehensively evaluate the effectiveness of the enhanced algorithms proposed in this study under different experimental scenarios, two distinct platforms were utilized: a PC development environment, and an embedded development environment. This approach allowed for a comprehensive assessment of the performance of the proposed algorithm across diverse computing environments. The Windows 10 × 64 operating system was selected for the PC development platform, with an Intel^®^ (Santa Clara, CA, USA) Core™ i7-10700F CPU 2.90 GHz, NVidia GeForce RTX 3070 8 GB GPU, and 32.0 GB RAM of running memory.

The embedded experimental platform used the NVidia Jetson Nano (NVIDIA: Santa Clara, CA, USA) device for model inference and testing. The experimental environment was Ubuntu 18.04 with Jetpack 4.5, CUDA 10.2, and cuDNN 8.0. The programming language was Python 3.6, and the deep learning framework included Pytorch 1.8.1 and Torchvision 0.9.1. Real-time detection using the Jetson Nano is shown in Figure 9.

In the PC platform, the training-specific parameters were as follows: the input image was 640 × 640 pixels, the batch size was 8, the initial learning rate was set to 0.001, and the optimizer was set to Adam. The number of training iterations was set to 70 to obtain a better model, and we applied loss and AP value change curves to the test set after 70 training sessions, as shown in Figure 10. During the first 15 cycles of network training, the loss value of the network decreased rapidly and the AP value increased rapidly, after which both entered stable convergence phases. After 60 epochs, the loss value decreased gently, the AP value increased gently, and the loss function curve and AP value curve converged, indicating that the model training effect was successful.

### 3.2. Evaluation Metrics

To assess the performance of the improved YOLOv5s model based on its detection results, we employed several evaluation metrics, including precision (*P*), recall (*R*), average precision (*AP*), number of parameters, floating point operations per second (*FLOPs*), model size, and frames per second (*FPS*). Taking passion fruit samples as an example, precision refers to the proportion of correctly predicted passion fruit samples to all predicted passion fruit samples by the model classifier, while recall represents the proportion of correctly predicted passion fruit samples to the actual positive passion fruit samples. These are shown in Equations (7) and (8), respectively. However, Precision and Recall do not allow for direct assessment of detection accuracy. The performance of the detection network is assessed using the average precision (*AP*), which represents the average accuracy in detection, as defined in Equation (Equation 9). In these equations, *TP* represents actual passion fruits predicted to be passion fruits, *FP* represents instances that are not actually passion fruits predicted to be passion fruits, and *FN* represents actual passion fruits predicted as not passion fruits. The number of floating point operations per second reflects the time complexity of the model, measuring the computation involved in operations such as convolution and pooling. On the other hand, the number of parameters describes the size of the model and its spatial complexity in the algorithm. Finally, the frames per second is used to measure the real-time performance of the model on the hardware platform.
(7)$$P=\frac{TP}{TP+FP}$$
(8)$$R=\frac{TP}{TP+FN}$$
(9)$$AP=\int_{0}^{1}P\left(R\right)dR$$

## 4. Experimental Results and Analysis

### 4.1. Impact of Different Backbone Networks on the Algorithm

In order to select a network with better lightweight performance as the backbone network for the YOLOv5s model, the first experiment compared the performance impact of the lightweight network layers and configurations on the backbone network. For this purpose, three types of backbone networks were selected: MobileNetv3, ShuffleNetv3, and GhostNet. To avoid reasonable bias, the FPN network and detection head were kept constant. The experiment was conducted for a state-of-the-art comparison on the embedded platform Jetson Nano, and the results are presented in Table 3.

Compared to the original YOLOv5s model, the improved model (G-YOLOv5) has an average precision (AP) reduction of 2.30%. The reason for this is that the reduced number of model parameters and convolutional layers of the G-YOLOv5 model led to a reduction in the network’s ability to extract features compared to the M-YOLOv5 and S-YOLOv5 models, with respective AP improvements of 0.6% and 0.2%. Meanwhile, the network model volume was 7.10 MB, a reduction by 50.69% compared to the original network. The FLOPs and number of parameters of the improved model were significantly reduced compared to the YOLOv5s model, with the FLOPs reduced by 58.86% and the parameters by 54.93%. Interestingly, there was a discrepancy between the results of the three networks after replacing the backbone and the results for the original network, which indicates that the effectiveness of the network is influenced by its total number of parameters and the particular network structure. Ultimately, reduced model weight was achieved through replacement of the backbone network.

### 4.2. Ablation Experiments

The ablation experiment focused on analyzing the value of each of the components in the improved method, and was conducted on the self-made dataset constructed in this study. The experimental results for the improved model are shown in Table 4.

By replacing the lightweight model, reconstructing the neck, and introducing knowledge distillation enhancement in the YOLOv5s baseline network, the AP of the improved network was able to meet the detection requirements. At the same time, the precision, recall, and AP values of the model all declined. To compensate for the loss of accuracy caused by replacing the backbone network, we reconstructed the neck network to pass useful information from the redundant feature layer to the neck network for fusion, where the size of the passion fruit targets in the dataset was also taken into account. This approach fuses the low-level semantic information of passion fruit targets with the high-level location information to obtain more useful feature layers, leading to an increase in AP from 93.10% to 93.60%, an improvement of about 0.5%. Finally, using the knowledge distillation learning approach, in which the teacher model passes rich information about the passion fruit features on to the student model, led to a substantial improvement in the average accuracy of the model. Compared to the original YOLOv5s model, the improved model had a mean average precision improvement of 1.00% and a size reduction of 50.42%.

### 4.3. Effect of Different Temperatures on the Algorithm

In our distillation experiments, we found that the temperature coefficient has a significant effect on the distillation effect. Therefore, the effect of different temperature coefficients on knowledge distillation results was explored on the basis of the student model (G-YOLOv5-N) and the teacher model (YOLOv5x). The specific approach that we employed was to select different temperature coefficients sequentially during the distillation experiments while maintaining a weighting factor of 0.5 in order to achieve a balance between the knowledge distillation loss and the losses of the original network. The knowledge distillation results at different temperatures are shown in Figure 11.

When the temperature coefficient was 20, the distilled model G-YOLO-NK had high recall and average precision results of 92.10% and 96.40%, respectively. With different distillation temperature coefficients, the accuracy and recall curves fluctuated up and down, indicating that different temperature coefficients cause the model to focus on different characteristics of the passion fruit targets. The average precision tended to be higher, indicating that the distilled model performs well at identifying passion fruits in complex environments.

### 4.4. Comparison with State-of-the-Art Models

To compare the performance of the improved model with current mainstream target detection models, we tested SSD, Faster R-CNN, RetinaNet, YOLOv5s, YOLOv5x, YOLOv6, YOLOv7-tiny, and YOLOv8s on the Jetson Nano. The comparison involved indicators such as floating point operations per second (FLOPs), number of parameters, frames per second (FPS), precision, recall, average precision (AP), and model size, all on the same self-made dataset. Additionally, we compared the method proposed in this article with recent research focused on passion fruit detection. It is important to note that focusing solely on a single evaluation indicator on different datasets and platforms is unfair; thus, we discuss similar research methods with the same research targets. The relevant performance indicators are presented in Table 5, which is incomplete, as some data were not provided.

Due to the instability of the frame rate during real-time detection with the Jetson Nano platform, the frame rate in this experiment was the average of 100 detected frame rates. The improved AP for the G-YOLO-NK model was 96.40%, which was higher by 15.41%, 6.50%, 1.51%, 1.00%, 0.30%, 7.13%, 6.20%, and 1.00%, respectively, compared to the SSD, Faster R-CNN, RetinaNet, YOLOv5s, YOLOv5x, YOLOv6s, YOLOv7-tiny, and YOLOv8s models. Obviously, the G-YOLO-NK model had better Average Precision (AP), indicating that the model is capable of detecting passion fruits in complex environments. In terms of real-time detection speed, the G-YOLO-NK model had better real-time detection rate on both the PC and Jetson Nano, with 125.00 f/s and 11.23 f/s, respectively. Compared to the various YOLOv5, YOLOv6, YOLOv7-tiny, and YOLOv8s models, it achieved average frame rate improvements ranging from 3.30 to 10.95 f/s on the Jetson Nano. The size, FLOPs, and number of parameters of improved model were 7.14 MB, 6.60 G, and 3.51 M, respectively, representing reductions of 50.42%, 59.51%, and 50.56% with respect to the YOLOv5s model, further proving the effectiveness and superiority of the improved network. In summary, the G-YOLO-NK model outperformed the extant models for detecting passion fruit in all metrics and presented good overall performance, making it the most promising model for high-performance real-time passion fruit detection.

Comparing the existing methods presented in Table 5, Wu et al. [44] and Tang et al. [45] improved the detection accuracy of YOLOv3 by integrating DenseNet and reducing multiscale prediction. Tu et al. [46] enhanced the network’s extraction ability by incorporating an FPN into Faster R-CNN. Furthermore, Tu et al. [47] fused RGB and depth color spaces in an improved Faster R-CNN to improve the model’s recognition rate. However, these studies did not address the need for light model weight and comprehensive deployment capability on mobile devices. Luo et al. [48] replaced the backbone network with MobileNet and integrated an attention mechanism to make the YOLOv5s network lighter; however, the average accuracy was low. Ou et al. [49] used ShuffleOne as the backbone network of YOLOv7 to reduce the number of network parameters and added a SimSPPF network at the neck to enhance the model’s fusion ability; however, the approach required a significant amount of time. Comparing the results of the above methods, G-YOLO-NK achieves higher AP and FPS than YOLOv3 proposed by Wu et al. (2020) [44]. The improved model proposed by Tang et al. (2020) [45] had an average accuracy 1.1% higher than that of G-YOLO-NK, but the detection speed of G-YOLO-NK was more than three times that of YOLOv3. Overall, G-YOLO-NK is a smaller model than the Faster R-CNN models proposed by Tu et al. (2020) [46] and Tu et al. (2020) [47]. Furthermore, when comparing detection frame rates on the PC, G-YOLO-NK demonstrated higher detection speeds than the other recent models.

To compare the generalization ability of the proposed model, we evaluated its performance on the COCO128 and VOC2012 datasets. Both datasets were randomly divided into training and testing sets using an 8:2 ratio. The results on the testing sets are shown in Table 6. On the COCO128 dataset, G-YOLO-NK improved the precision by 0.2%, recall by 0.9%, and average precision by 0.8% compared to YOLOv5. On the VOC2012 dataset, it improved recall by 0.8% and average precision by 0.5%, compared to YOLOv5 while reducing precision by 0.2%. This lower precision may be due to the lack of an accurate learning rate setting; however, this does not affect the overall model evaluation. In summary, the comparison between the two datasets demonstrates that G-YOLO-NK has high generalization ability and strong robustness.

AI approaches require efficient computing power to process large amounts of data. GPUs play a crucial role in this process due to their numerous cores, high-speed memory, and parallel computing capabilities, which significantly alleviate computational bottlenecks and make deep learning algorithms practical. GPU utilization serves as an indicator of resource utilization on the GPU. Excessive GPU usage can lead to freezing and crashes during real-time target detection on devices such as the Jetson Nano. Prolonged high GPU usage can also impact the performance and lifespan of the system. Therefore, visualizing GPU occupancy is essential. When running the YOLOv5s and G-YOLO-NK models on the Jetson Nano for real-time passion fruit detection, we observed how GPU usage changed over time to ensure optimal performance and longevity of the device. The visualization results for the two models are shown in Figure 12.

As can be seen from Figure 12, the GPU occupancy rates of the two models running on the Jetson Nano differ. Greater network complexityis indicated by a denser blue color in the bar graph, indicating higher demand for GPU resources on the embedded device. The G-YOLO-NK model requires the least computation for real-time detection on the Jetson Nano, underscoring the importance of lightweight model design and improved efficiency.

### 4.5. Comparison of Recognition Effect before and after Improvement

To verify the detection performance of the G-YOLO-NK model, passion fruit images captured in complex environments, including dense, shaded, sunny, and rainy conditions, were selected for comparison testing against the original model. A confidence threshold of 0.7 and an IoU threshold of 0.5 were chosen. The detection results of the YOLOv5s and G-YOLO-NK models on embedded devices are shown in Figure 13.

In the figure, the blue rectangular boxes are predicted target boxes and the red rectangular boxes are missed targets. Both types of models correctly detected the passion fruit targets in backlit, overcast, and rainy weather scenes. The YOLOv5s model missed passion fruit detections in both dense and shaded situations, while the G-YOLO-NK model correctly plotted the predicted boxes. The YOLOv5s model missed two passion fruits in the sunny image, while G-YOLO-NK missed one passion fruit, as the targets were too small. In terms of confidence, G-YOLO-NK had a higher confidence level than the YOLOv5s model, indicating the good detection effect of the improved model and the effectiveness of the improved method. In summary, it can be concluded from the recognition results that the G-YOLO-NK model provides improved detection performance under complex environmental conditions, showing good robustness and generalization ability.

Visualizing the feature layer illustrates the performance of model feature extraction and the distribution of contributions to the predicted output, providing a more representative analysis. Using the shaded passion fruit image as an example, the overall feature maps for the deep convolutional layers of YOLOv5s, YOLOv7 tiny, and YOLOv8s were compared against that of the proposed model, with the results shown in Figure 14. It can be seen that the feature maps of YOLOv5s, YOLOv7 tiny, and YOLOv8s each include six highlighted areas, while that of G-YOLO-NK has eight highlighted areas. Each highlight corresponds to a passion fruit in the original image. This indicates that the G-YOLO-NK model has superior feature extraction ability and more accurate prediction ability.

## 5. Conclusions

In this study, a lightweight and high-precision target detection model based on G-YOLO-NK is proposed for identification of passion fruits. The YOLOv5s network model was optimized by replacing the backbone network, reconstructing the neck feature fusion network, and introducing knowledge distillation, resulting in greatly improved detection accuracy and speed in complex environments. The model’s size and number of parameters were drastically reduced, which is conducive to deploying the detection model on constrained mobile terminals with limited computing power. The proposed model achieved a frame rate of 11.23 FPS for passion fruit detection on the Jetson Nano, as well as a precision of 93.00%, recall of 92.1%, and average precision of 96.4%. Compared with the original YOLOv5, the proposed network was found to have improved accuracy (by 1.0%) and reduced model size (by 50.4%). The proposed model significantly outperformed state-of-the-art models in terms of detection performance and accuracy in scenes characterized by complex conditions. For this study, a passion fruit dataset was collected under different location, lighting, and weather conditions and expanded through the use of several data augmentation methods, resulting in improved robustness on the part of the model. It should be noted that the G-YOLO-NK model has a number of limitations; for example, missed detections can occur when leaves or fruits occlude more than 90% of the passion fruit area. In the future, we will use segmentation technology and RGB-D fusion technology to improve the missed detection rate. In addition, we plan to collect additional passion fruit datasets that include varying ripening stages and different colors to enable multi-class detection of passion fruits. Finally, the model proposed in this study could be applied to estimate the yield of passion fruit and provide assistance for predicting other fruits in the field.

## Figures and Tables

**Figure 1 sensors-24-04942-f001:**
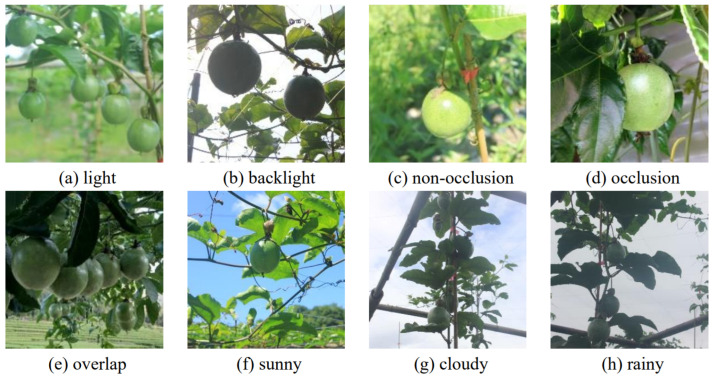
Pictures of passion fruit in complex environments.

**Figure 2 sensors-24-04942-f002:**
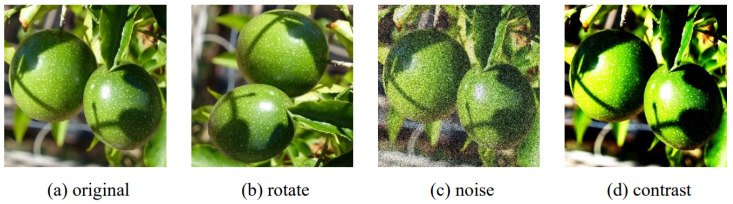
Data augmentation.

**Figure 3 sensors-24-04942-f003:**
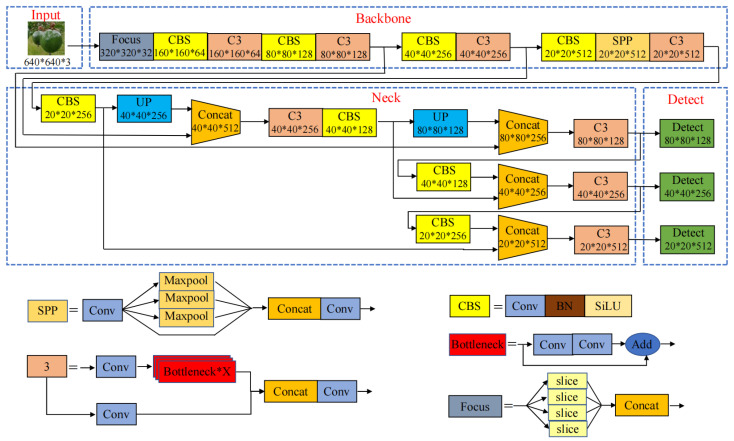
Architectureof the Yolov5 algorithm.

**Figure 4 sensors-24-04942-f004:**
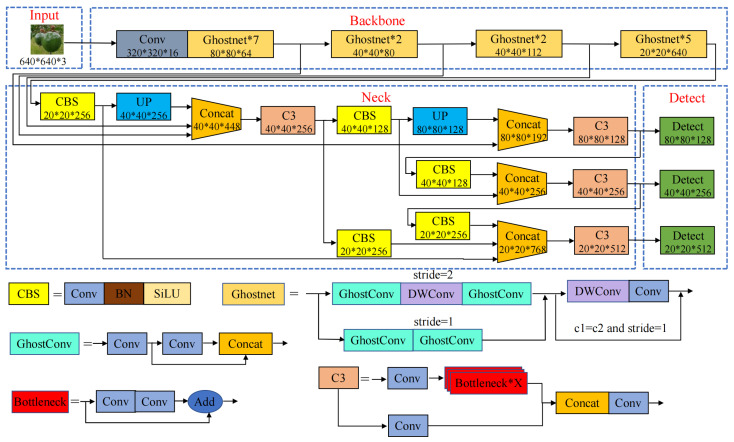
Structure of the G-YOLO-N algorithm.

**Figure 5 sensors-24-04942-f005:**
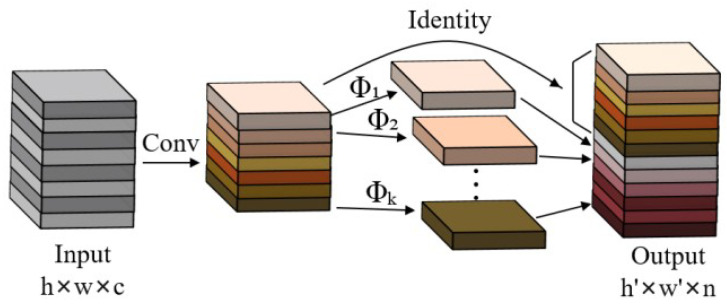
Structure diagram of the Ghost module.

**Figure 6 sensors-24-04942-f006:**
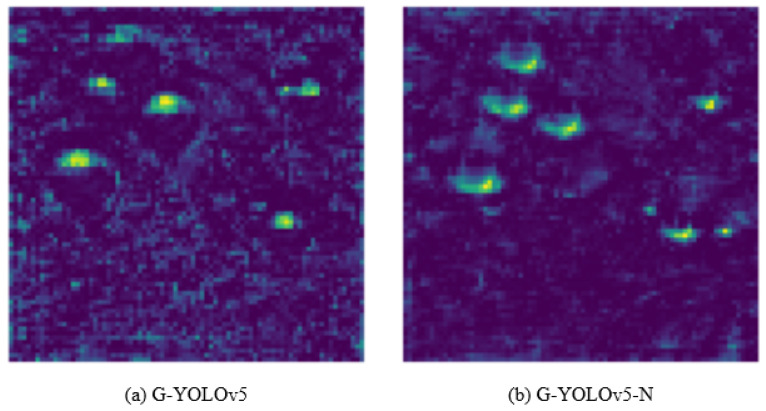
Improved neck network: before and after.

**Figure 7 sensors-24-04942-f007:**
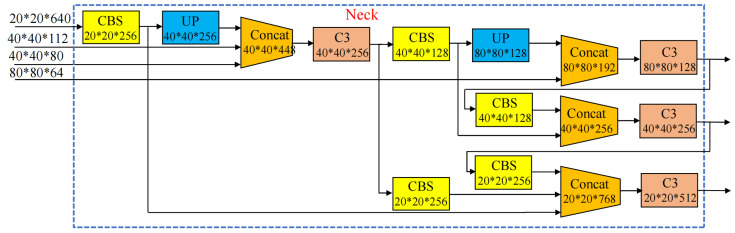
Improved neck network structure.

**Figure 8 sensors-24-04942-f008:**
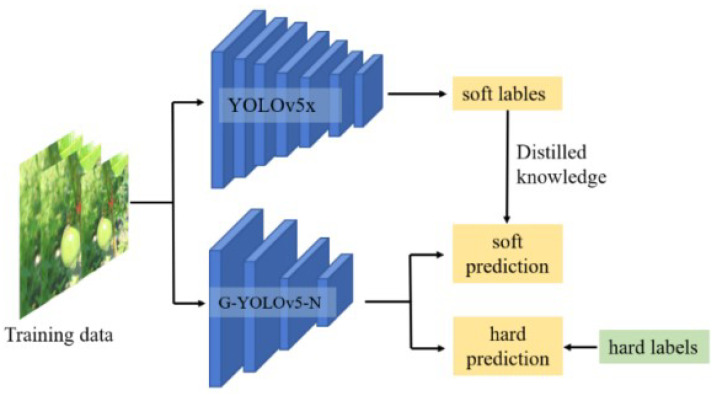
Flowchart of the knowledge distillation algorithm.

**Figure 9 sensors-24-04942-f009:**
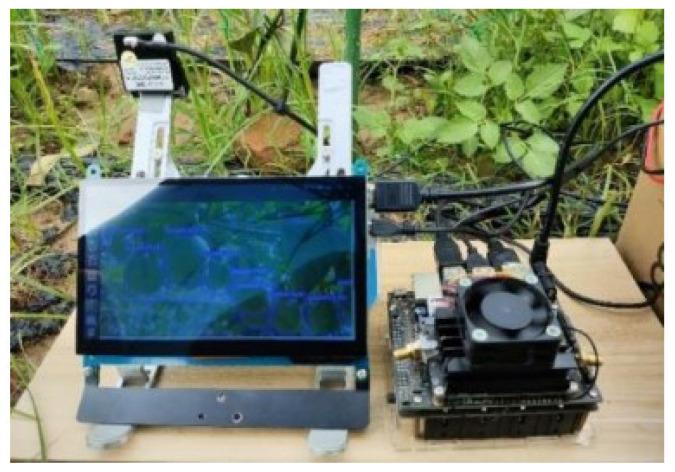
The real-time detection environment using Jetson Nano.

**Figure 10 sensors-24-04942-f010:**
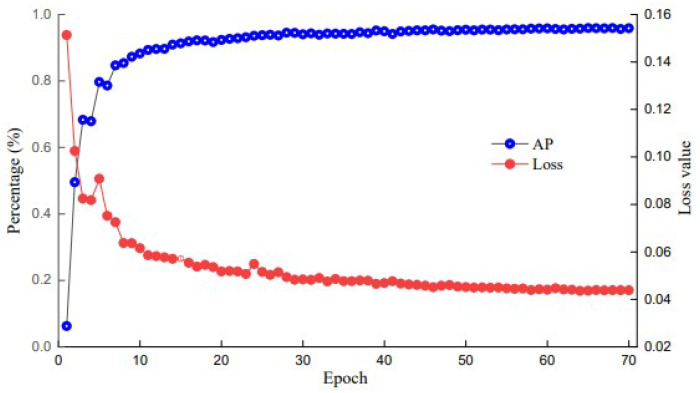
AP and loss curves.

**Figure 11 sensors-24-04942-f011:**
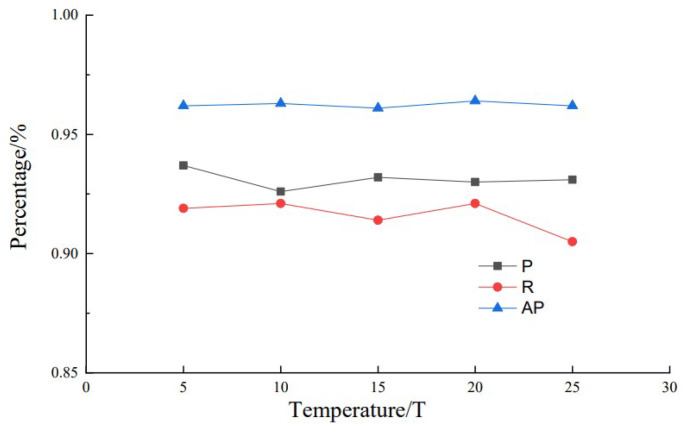
Indicators at different temperatures.

**Figure 12 sensors-24-04942-f012:**
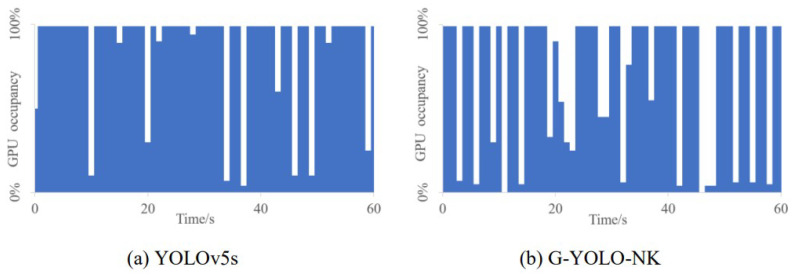
Visualization of GPU utilization.

**Figure 13 sensors-24-04942-f013:**
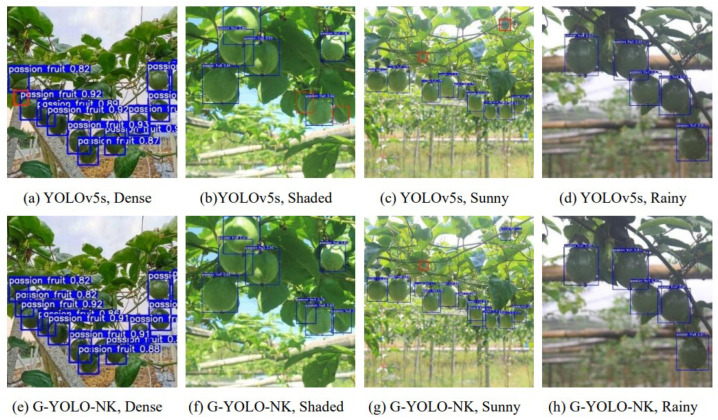
Comparison of detection results between the YOLOv5s and G-YOLO-NK models.

**Figure 14 sensors-24-04942-f014:**
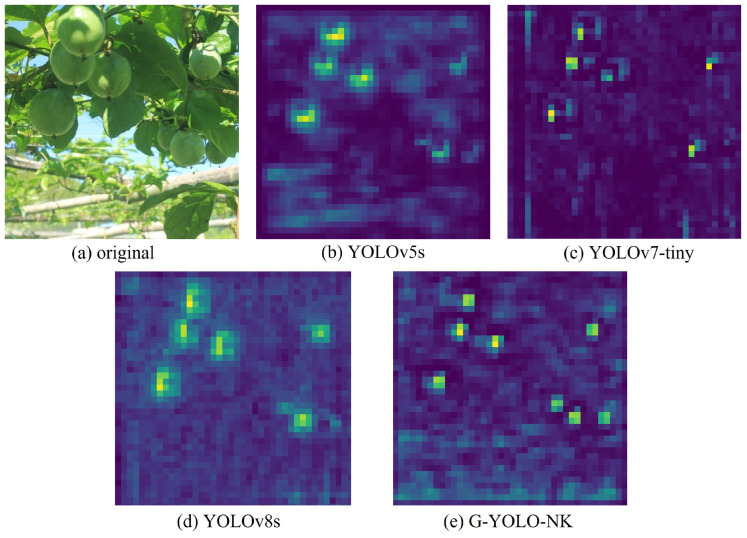
Comparison of feature detection layers.

**Table 1 sensors-24-04942-t001:** Performance comparison of different YOLOv5 models.

Model	P (%)	R (%)	AP (%)	Size (MB)
YOLOv5s	94.90	90.60	95.40	14.40
YOLOv5m	94.80	90.90	95.70	40.10
YOLOv5l	94.80	91.20	96.00	88.40
YOLOv5x	94.90	92.20	96.10	164.00

**Table 2 sensors-24-04942-t002:** Overall structure of GhostNet as the backbone network.

Layer	Operator	Out	Stride	Layer	Operator	Out	Stride
1	Conv2d	16	2	10	GhostNet	80	1
2	GhostNet	16	1	11	GhostNet	112	2
3	GhostNet	24	2	12	GhostNet	112	1
4	GhostNet	24	1	13	GhostNet	160	2
5	GhostNet	40	1	14	GhostNet	160	1
6	GhostNet	40	1	15	GhostNet	320	1
7	GhostNet	64	2	16	GhostNet	320	1
8	GhostNet	64	1	17	GhostNet	640	1
9	GhostNet	80	2				

**Table 3 sensors-24-04942-t003:** Comparison of different backbone networks.

Model	P (%)	R (%)	AP (%)	GFLOPs	Param (M)	Size (MB)
YOLOv5s	94.90	90.60	95.40	15.80	7.10	14.40
M-YOLOv5	92.50	86.60	92.50	6.30	3.54	7.08
S-YOLOv5	92.50	87.30	92.90	7.40	3.55	7.12
G-YOLOv5	92.90	87.40	93.10	6.50	3.20	7.10

**Table 4 sensors-24-04942-t004:** Results of the ablation experiment.

Baseline	Light	Neck	KD	P (%)	R (%)	AP (%)	Size (MB)
YOLOv5s				94.90	90.60	95.40	14.40
	✓			92.90	87.40	93.10	7.10
		✓		95.10	90.50	95.50	14.46
			✓	93.70	92.90	96.10	14.40
	✓	✓		93.70	87.40	93.60	7.14
	✓		✓	93.50	90.70	95.80	7.10
	✓	✓	✓	93.00	92.10	96.40	7.14

**Table 5 sensors-24-04942-t005:** Comparison with different models in recent works.

Model	Input	GFLOPs	Param	FPS	PC FPS	P (%)	R (%)	AP (%)	Size	Literature	Number of Images
SSD	512 × 512	61.20	100.10	0.79	5.60	85.48	80.26	80.99	90.60	\	5140
Faster R-CNN	600 × 600	273.40	118.20	0.28	2.96	90.54	87.80	89.90	521.00	\	5140
RetinaNet	512 × 512	145.51	36.39	0.58	4.94	75.49	96.00	94.89	138.00	\	5140
YOLOv5s	640 × 640	16.30	7.10	6.19	78.74	94.90	90.60	95.40	14.40	\	5140
YOLOv5x	640 × 640	203.80	86.17	0.63	50.00	94.90	92.20	96.10	173.21	\	5140
YOLOv6s	640 × 640	45.17	18.50	3.10	76.00	75.40	81.20	89.27	38.70	\	5140
YOLOv7-tiny	640 × 640	13.00	6.01	7.93	90.14	92.30	89.90	90.20	11.60	\	5140
YOLOv8s	640 × 640	28.40	11.13	3.98	83.33	95.70	92.30	95.40	21.40	\	5140
G-YOLO-NK	640 × 640	6.60	3.51	11.23	125.00	93.00	92.10	96.40	7.14	\	5140
YOLOv3	640 × 640	\	\	\	36.00	\	\	86.70	\	Wu et al., 2020 [44]	1000
YOLOv3	640 × 640	\	\	\	38.00	97.60	96.50	97.50	\	Tang et al., 2020 [45]	4000
Faster R-CNN	600 × 600	\	\	\	5.71	96.20	93.10	\	\	Tu et al., 2020 [46]	8651
Faster R-CNN	600 × 600	\	\	\	5.62	90.79	90.47	\	\	Tu et al., 2021 [47]	2275
YOLOv5	640 × 640	6.60	\	10.92	124.70	95.30	88.10	88.30	6.41	Luo et al., 2022 [48]	2000
YOLOv7	640 × 640	\	\	\	58.20	91.20	\	90.45	29.20	Ou et al., 2023 [49]	3962

**Table 6 sensors-24-04942-t006:** Comparison of models on different datasets.

Model	Input	Dataset	P (%)	R (%)	AP (%)	Size (MB)
YOLOv5	640 × 640	COCO128	88.30	83.00	89.90	14.40
G-YOLO-NK	640 × 640	COCO128	88.50	83.90	90.70	7.14
YOLOv5	640 × 640	VOC2012	69.60	62.70	67.10	14.40
G-YOLO-NK	640 × 640	VOC2012	69.40	63.50	67.60	7.14

## Data Availability

All data are contained within the article. To request the data and code, please send an email to the first or corresponding author.

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
