# Peer review of "A Lightweight and High-Precision Passion Fruit YOLO Detection Model for Deployment in Embedded Devices"

_sensors, 2024, doi:10.3390/s24154942_

Round 1

Reviewer 1 Report

Comments and Suggestions for Authors

1. The method examined passion fruit under different environments or light, but the effect of ripeness of passion fruit was not mentioned. The authors should increase the discussion of this problem.

2. YOLO series algorithms have been iterated many times in recent years, and many versions with better performance have appeared. Why choose YOLOv5 as the basic algorithm? For example, YOLOv8 has obvious performance advantages, why not improve on the basis of this algorithm?

3. The structure diagram of YOLOv5 algorithm is highly similar to the structure diagram of G-YOLO-N algorithm, and there is no outstanding improvement.

4. In Table 4, why are the input images of the first three networks different in size?

Comments on the Quality of English Language

Extensive editing of English language required.

Author Response

Comments 1: The method examined passion fruit under different environments or light, but the effect of ripeness of passion fruit was not mentioned. The authors should increase the discussion of this problem.

Response 1:

Thank you very much for your suggestions.We acknowledge that the detection of passion fruit ripeness is a significant research direction. However, in this paper, our primary focus is on the detection of unripe passion fruit targets. Our objective is to develop a method capable of accurately detecting unripe passion fruit under various environmental and lighting conditions, thereby laying the foundation for predicting the yield of passion fruits.In the conclusion, we have made a prospect for the detection of passion fruit maturity.

On page 15, line 444 of the conclusion section of the revised manuscript.

Comments 2: YOLO series algorithms have been iterated many times in recent years, and many versions with better performance have appeared. Why choose YOLOv5 as the basic algorithm? For example, YOLOv8 has obvious performance advantages, why not improve on the basis of this algorithm?

Response 2: 

Thank you very much for your suggestions. We have tested our model on our own dataset,and the results are shown in Table 5 of the reversed manuscript.

Compared to YOLOv5, although YOLOv8 has higher precision (P), it also comes with greater computational complexity (GFLOPs). When performing detection on embedded devices, it is essential to consider the overall model factors, and YOLOv8 is less suitable for real-time detection on such devices. Therefore, we chose YOLOv5 as the baseline network for our improvements.

Moreover, in the revised manuscript, Table 5 presents the test results on our own dataset. Despite YOLOv5 and YOLOv8 achieving the same average precision (AP), YOLOv5 has fewer parameters and a higher real-time detection frame rate. We believe that the reduction in the number of parameters and the increase in frame rate are crucial for practical applications. Fewer parameters mean a more lightweight model, which is easier to deploy on resource-constrained devices. Additionally, a higher frame rate ensures that the model can provide fast and accurate detection results in applications requiring real-time feedback.

Comments 3:The structure diagram of YOLOv5 algorithm is highly similar to the structure diagram of G-YOLO-N algorithm, and there is no outstanding improvement.

Response 3: 

Thank you for pointing this out.Regarding your concern about the similarity between the YOLOv5 algorithm and the G-YOLO-N algorithm without significant improvements, I would like to provide a detailed explanation and response here. From the structural diagrams, YOLOv5 and G-YOLO-N share some similarities in overall architecture. To clearly demonstrate the uniqueness of G-YOLO-N, we have made revisions to Figures 3, 4, and 6 in the revised manuscript, as shown below.

Comments 4:In Table 4, why are the input images of the first three networks different in size?

Response 4:

Thank you very much for your suggestions. We would like to provide the following explanation: When selecting the input image sizes for different networks, we primarily referred to previous studies published by scholars and chose corresponding input sizes accordingly. This choice helps us assess the performance of these networks more accurately on our dataset and facilitates comparison with prior research.

Reviewer 2 Report

Comments and Suggestions for Authors

This paper constructs a passion fruit dataset and proposes a lightweight passion fruit target detection algorithm. The algorithm effectively improves the detection speed and accuracy of passion fruit detection. Here are my comments for further improving the manuscript:

  1. There should be a full stop before the sentence "In the sentence" in line 7 in the Introduction section on the first page.
  2. Italics are recommended for parameters in the manuscript.
  3. Correct the typo “the YOLOv5x model with hight accuracy” to “...high accuracy.”

Additional Comments:

  1. The language formatting of the article is inconsistent. For example, the presentation of figures in section 2.1 is not consistent. Additionally, the article is heavily colloquial; revisions are recommended to improve readability.
  2. The images in this paper are distorted when zoomed in. It is recommended to use vector images instead.
  3. This paper mentions that GhostNet outperforms MobileNet and ShuffleNet in terms of computational performance, leading to its selection. It is recommended to add experiments for verification. Additionally, the extensive introduction to GhostNet is unnecessary as the authors have not made improvements to it.
  4. The authors chose YOLOv5x, the largest and best performing of the YOLOv5 models, as the teacher model. This raises concerns that G-YOLOv5-N's performance is high mainly due to YOLOv5x's capabilities.
  5. The authors conducted experiments only on their own dataset, which has not been made public. This reduces the reliability of the results. It is recommended to add more comparative experiments using other public datasets.
  6. There are too few comparative models, and comparisons of recent works are lacking. Additionally, the comparison visualization is unconvincing as it only compares with YOLOv5s.
  7. Please standardize the reference format.
Comments on the Quality of English Language

Proper proofreading is required for this paper.

Author Response

Comments 1: There should be a full stop before the sentence "In the sentence" in line 7 in the Introduction section on the first page.

Response 1:

Thank you very much for your suggestions. We apologize for any punctuation and grammar issues that may have existed in the initial draft. We have carefully reviewed the entire paper and made necessary corrections to punctuation marks to ensure the correctness and clarity of sentences throughout the manuscript.

Comments 2: Italics are recommended for parameters in the manuscript.

Response 2: 

Thank you for your meticulous review of our paper. Your suggestion to use italics for parameters in the manuscript is a very useful formatting recommendation. We will follow your advice and format all parameters in the manuscript in italics to ensure consistency.

Comments 3: Correct the typo “the YOLOv5x model with hight accuracy” to “...high accuracy.”

Response 3: 

Thank you very much for your thorough review. We sincerely apologize for the typographical error found in the manuscript. We will promptly correct "hight" to "high" to ensure the accuracy of the sentence. To prevent similar errors, we have proofread the entire paper to ensure all text is accurate.

Comments 4: The language formatting of the article is inconsistent. For example, the presentation of figures in section 2.1 is not consistent. Additionally, the article is heavily colloquial; revisions are recommended to improve readability.

Response 4:

Thank you for your thorough review of our paper and valuable suggestions. We will conduct a formatting review of all figures and illustrations to ensure consistent presentation throughout the manuscript. Additionally, we will perform a comprehensive grammar check on the manuscript, adopting a more academic and formal writing style to further enhance the quality and readability of the article.

Comments 5: The images in this paper are distorted when zoomed in. It is recommended to use vector images instead.

Response 5:

Thank you for your meticulous review of our paper. We highly value your feedback regarding the image quality issue. Therefore, we have redrawn the images in the manuscript using professional graphics software to ensure they do not distort when enlarged.

Comments 6: This paper mentions that GhostNet outperforms MobileNet and ShuffleNet in terms of computational performance, leading to its selection. It is recommended to add experiments for verification. Additionally, the extensive introduction to GhostNet is unnecessary as the authors have not made improvements to it.

Response 6:

We appreciate your thorough review and valuable suggestions on our paper. Regarding your concerns about validating the performance of GhostNet and the scope of its introduction, we have made the following responses and adjustments: We designed corresponding experiments to compare the performance of GhostNet, MobileNet, and ShuffleNet under identical hardware conditions.

The experimental results are shown in the table below:

Furthermore, we understand that extensive introduction of GhostNet without improvements may not be necessary. Therefore, we have simplified the content of Section 2.3.1 "Lightweight improvements," retaining only information relevant to our study.

Comments 7: The authors chose YOLOv5x, the largest and best performing of the YOLOv5 models, as the teacher model. This raises concerns that G-YOLOv5-N's performance is high mainly due to YOLOv5x's capabilities.

Thank you for your concern regarding our choice of YOLOv5x as the teacher model and for your valuable feedback. We understand your apprehension that the high performance of G-YOLOv5-N may primarily be attributed to the inherent capabilities of YOLOv5x. Here is our detailed response to this issue:

Before improving the model, we established two core objectives: first, to achieve model lightweighting to meet the operational requirements of embedded devices; second, to enhance recognition accuracy without increasing the model size. To address the latter, we improved the model using knowledge distillation methods. Our experimental results clearly demonstrate that the performance of G-YOLOv5-N has been substantially improved through knowledge distillation. This improvement is not solely attributed to using YOLOv5x as the teacher model, but also to the carefully designed strategies employed during model training and distillation

Comments 8:The authors conducted experiments only on their own dataset, which has not been made public. This reduces the reliability of the results. It is recommended to add more comparative experiments using other public datasets.

Response 8:

Thank you for your valuable feedback. We understand your concern regarding the reliability of our experimental results. To increase the transparency and reproducibility of our research, we have publicly shared our dataset. We have uploaded the dataset to Baidu Netdisk and provided the corresponding download links to ensure accessibility. The dataset is openly available for download, allowing other researchers to verify our experimental results or use it for their own studies.

Link:

https://pan.baidu.com/s/1jxEF3hXXHaiB52nC-W3Hxg

code:frc4

Comments 9: There are too few comparative models, and comparisons of recent works are lacking. Additionally, the comparison visualization is unconvincing as it only compares with YOLOv5s.

Response 9:

Thank you for your suggestion.Compare recent works in the first paragraph of page 13 of the revised manuscript.and present them in the form of a table.

Secondly, we have added visualizations and analyses of YOLOv8 and YOLOv7 models in the revised manuscript.

Comments 10:Please standardize the reference format.

Response 10:

Thank you for your suggestion. We have proofread the format of all references to ensure they comply with academic standards.

Reviewer 3 Report

Comments and Suggestions for Authors

In this paper, the authors consider a lightweight and high accuracy model for passion fruit in complex environments (backlight, occlusion, overlap, sunny, cloudy, rainy). A new feature branch is added to the GhostNet network, and the feature fusion layer in the neck network is reconstructed to effectively combine the lower-level and higher-level features, which not only improves the accuracy of the model but also maintains its lightweight. The topic is quite interesting and this paper is well-written. I have the following minor comments for the further improvement of the paper.

1) The description of the Abstract part is too lengthy, it is recommended to simplify.

2) The authors are suggested to present a pseudocode or procedure for the used model in the paper.

3) It is suggested that the author explain the whole testing equipment and experimental environment setting clearly with block diagram and text.

4) The authors are suggested to present more future potential works in the conclusion part.

Comments on the Quality of English Language

In this paper, the authors consider a lightweight and high accuracy model for passion fruit in complex environments (backlight, occlusion, overlap, sunny, cloudy, rainy). A new feature branch is added to the GhostNet network, and the feature fusion layer in the neck network is reconstructed to effectively combine the lower-level and higher-level features, which not only improves the accuracy of the model but also maintains its lightweight. The topic is quite interesting and this paper is well-written. I have the following minor comments for the further improvement of the paper.

1) The description of the Abstract part is too lengthy, it is recommended to simplify.

2) The authors are suggested to present a pseudocode or procedure for the used model in the paper.

3) It is suggested that the author explain the whole testing equipment and experimental environment setting clearly with block diagram and text.

4) The authors are suggested to present more future potential works in the conclusion part.

Author Response

Comments 1: The description of the Abstract part is too lengthy, it is recommended to simplify

Response 1:

We acknowledge that the detection of passion fruit ripeness is a significant research direction. However, in this paper, our primary focus is on the detection of unripe passion fruit targets. Our objective is to develop a method capable of accurately detecting unripe passion fruit under various environmental and lighting conditions, thereby laying the foundation for predicting the yield of passion fruits.In the conclusion, we have made a prospect for the detection of passion fruit maturity.

The revised abstract is as follows:

In order to shorten the detection time and improve the average precision on embedded devices, a lightweight and high accuracy model is proposed for detecting passion fruit in complex environments (backlight, occlusion, overlap, sunny, cloudy, rainy). Firstly, replacing the backbone network of YOLOv5 with a lightweight GhostNet model reduces the number of parameters and computation while improving detection speed. Secondly, a new feature branch is added to the GhostNet network, and the feature fusion layer in the neck network is reconstructed to effectively combine the lower-level and higher-level features, which not only improves the accuracy of the model but also maintains its lightweight. Finally, the knowledge distillation methods are used to transfer the knowledge from the more capable teacher model to the less capable student model,significantly improving the detection accuracy. The improved model is denoted as G-YOLO-NK. The average accuracy of the G-YOLO-NK network is 96.00$\%$, which is 1.00$\%$ higher than of the original YOLOv5s model. Furthermore, the improved model size is 7.14MB, reduced to half of the original model, and the real-time detection frame rate is 11.25 FPS on the Jetson Nano. Compared to the state-of-the-art model, the proposed model outperforms them in terms of average precision and detection performance. The present work provides an effective model for real-time detection of passion fruits in complex orchard scenes, offering valuable technical support for the development of orchard picking robots and greatly improve the intelligence level of orchards.

Comments 2: The authors are suggested to present a pseudocode or procedure for the used model in the paper.

Response 2: 

Thank you for your suggestion. We understand that providing pseudocode or the model usage process could be helpful for readers to understand our research methodology. However, considering that adding pseudocode may affect the overall structure and allocation of space, we have already provided detailed descriptions in the paper, elucidating the working principles and usage process of the model to ensure readers can comprehend our approach. We appreciate your understanding and are willing to discuss any other issues further.

Comments 3: It is suggested that the author explain the whole testing equipment and experimental environment setting clearly with block diagram and text.

Response 3: 

Thank you for your suggestion. In the revised third section, we have provided detailed explanations of every aspect of the experimental setup, including hardware configurations, software environments, and network structures. We have listed specifications of all hardware devices used, such as processors, memory, and storage devices. Additionally, we have elaborated on the operating system, programming language, library versions, and all key parameters and settings used in the experiments, such as learning rate, batch size, and training epochs, to ensure that other researchers can replicate the experiments in the same or similar environments.

On page 9 of the revised manuscript, line 248 of section 3.1 Experimental Environment.

Comments 4: The authors are suggested to present more future potential works in the conclusion part.

Thank you for your suggestion. We recognize the importance of proposing future potential research directions in the conclusion section to guide subsequent studies and demonstrate the broader significance of the research. Therefore, we have extensively discussed several potential research directions in the conclusion, including further optimization of the model, improvement of algorithms, and extended applications in various scenarios.

The revised conclusion is as follows:

In this paper, a lightweight and high-precision target detection model based on G-YOLO-NK was proposed to identify passion fruit. The YOLOv5s network model was optimized by replacing the backbone network, reconstructing the neck feature fusion network, and intruducting knowledge distillation, which greatly improves the detection accuracy and speed in complex environments. The model’s size and the number of parameters was drastically reduced, which is conducive to deploying the detection model in a mobile terminal with limited computing power. The proposed model achieves a frame rate of 11.23 FPS of the Jetson Nano for passion fruit detection, a P of 93.00\%, an R of 92.1\%, and an AP of 96.4\%. Compared with the original YOLOv5, the proposed network improves accuracy by 1.0\% and reduces model size by 50.4\%. The proposed model significantly outperforms state-of-the-art models in terms of detection performance and accuracy in conditions of complex scenes. In this study, a passion fruit dataset was collected under different locations, lighting, and weather conditions. And expanding the dataset through different data augmentation methods has improved the robustness of the model.The G-YOLO-NK model has limitations, for example, when leaves or fruits occlude more than 90\% of the passion fruit area, it can cause missed detections.In the future, we will use segmentation technology and RGB-D fusion technology to improve the missed detection rate.we plan to collect passion fruit datasets with varying ripening stages and different colors to enable multi-classification detection of passion fruit. Finally, this study can be applied to estimate the yield of passion fruit and provide assistance for predicting other fruits in the field.

Round 2

Reviewer 2 Report

Comments and Suggestions for Authors

Although the author made some changes that slightly improved the quality of the manuscript, the core issues remain unaddressed. The following concerns need to be addressed by the authors:

  1. In addition to making the self-made dataset publicly available, it is crucial to verify the proposed method on other publicly available datasets. The author needs to include at least two public datasets in the experimental section to compare performance.
  2. In the revised manuscript, the author added several comparison methods. Some of these methods, such as Tang et al., outperform the proposed method in most metrics. The author needs to analyze and highlight the advantages of the proposed method.
  3. The authors need to standardize the contrast of the methods presented in Table 5 and Table 6. It is recommended to compare them within the same table for consistency.
  4. There are may typos in the new additions, such as:

l  On Page 7: "At the sametime, From the images in the dataset, …" should be "At the same time, from the images in the dataset, …"

l  "Therefore, We first connect the …" should be "Therefore, we first connect the …"

l  On Page 15: There should be a space in front of "A."

l  "we plan to collect" should be "We plan to collect" with a space before "We."

Comments on the Quality of English Language

Authors should review and address writing issues throughout the manuscript to ensure readability and accuracy.

Author Response

Comments 1: In addition to making the self-made dataset publicly available, it is crucial to verify the proposed method on other publicly available datasets. The author needs to include at least two public datasets in the experimental section to compare performance.

Response 1:

Thank you very much for your suggestions. We have validated the generalization ability of the proposed model on the VOC2012 and COCO128 public datasets. Display the model results before and after improvement in Table 6 of the revised manuscript. In the third paragraph of page 13, we compare the model results and analyze their implications.

Comments 2: In the revised manuscript, the author added several comparison methods. Some of these methods, such as Tang et al., outperform the proposed method in most metrics. The author needs to analyze and highlight the advantages of the proposed method.

Response 2: 

Thank you very much for your suggestions. We compared the proposed method with recent work and analyzed its advantages. This viewpoint is presented in the second paragraph on page 13 of the revised manuscript.

Comments 3: The authors need to standardize the contrast of the methods presented in Table 5 and Table 6. It is recommended to compare them within the same table for consistency.

Response 3: Thank you very much for your suggestions.In order to maintain consistency, we compared recent research work with different models and integrated the contents of Table 6 into Table 5.

Comments 4: There are may typos in the new additions, such as:

On Page 7: "At the sametime, From the images in the dataset, …" should be "At the same time, from the images in the dataset, …"

"Therefore, We first connect the …" should be "Therefore, we first connect the …"

On Page 15: There should be a space in front of "A."

 "we plan to collect" should be "We plan to collect" with a space before "We."

Response 4:

Thank you very much for your suggestions.We have corrected formatting errors in the content and verified the entire revised manuscript.
